# Rethinking Optimal Verification Granularity for Compute-Efficient Test-Time Scaling

**Hao (Mark) Chen**[1]    **Guanxi Lu**[1]    **Yasuyuki Okoshi**[2]
**Zhiwen Mo**[1]    **Masato Motomura**[2]    **Hongxiang Fan**[1]
[1]Imperial College London, UK
[2]Institute of Science Tokyo, Japan
*{hao.chen20, guanxi.lu22, zhiwen.mo25, hongxiang.fan}@imperial.ac.uk*
*{okoshi.yasuyuki, motomura}@artic.iir.titech.ac.jp*

## Abstract

Test-time scaling (TTS) has proven effective in enhancing the reasoning capabilities of large language models (LLMs). Verification plays a key role in TTS, simultaneously influencing (1) reasoning performance and (2) compute efficiency, due to the quality and computational cost of verification. In this work, we challenge the conventional paradigms of verification, and make the first attempt toward systematically investigating the impact of verification granularity—that is, how frequently the verifier is invoked during generation, beyond verifying only the final output or individual generation steps. To this end, we introduce **Variable Granularity Search (*VG-Search*)**, a unified algorithm that generalizes beam search and Best-of-N sampling via a tunable granularity parameter $g$. Extensive experiments with *VG-Search* under varying compute budgets, generator-verifier configurations, and task attributes reveal that dynamically selecting $g$ can improve the compute efficiency and scaling behavior. Building on these findings, we propose adaptive *VG-Search* strategies that achieve accuracy gains of up to 3.1% over Beam Search and 3.6% over Best-of-N, while reducing FLOPs by over 52%. Our code is avaiblae at github.com/hmarkc/VG-Search.

## 1   Introduction

The past few years have witnessed the rapid advancement of large language models (LLMs), driven by scaling of model size and training data [6, 17, 28]. However, further training-time scaling is increasingly constrained by prohibitive computational costs and the limited availability of high-quality human-generated data [34]. **Test-time scaling (TTS)** [33, 5, 32] offers a promising alternative by enhancing performance through additional compute at inference time. TTS techniques generally fall into two categories: *internal scaling* [33, 25], which focuses on optimizing a single generation trajectory, and *sampling-based scaling* [5, 32, 35], which improves performance by exploring multiple candidate generations. These two approaches are orthogonal and can be combined to achieve higher performance [22]. Sampling-based methods are typically training-free and easy to integrate with internal scaling approaches in a plug-and-play manner.

**Verification**, commonly implemented as a learned reward model or scoring function, plays a key role in both TTS paradigms. State-of-the-art (SOTA) sampling-based methods, such as Diverse Verifier Tree Search (DVTS) [4] and verifier-guided Beam Search [32], leverage a separate verifier LLM to guide the generation of a generator LLM, improving sampling efficiency and accuracy. In these methods, a **generation step** is typically defined as a chunk of text delimited by special tokens (e.g., newline characters), which serves as the atomic unit of verification. However, this verification granularity choice is heuristic [32, 22] and remains static [36], with no guarantee of optimality. As

39th Conference on Neural Information Processing Systems (NeurIPS 2025).

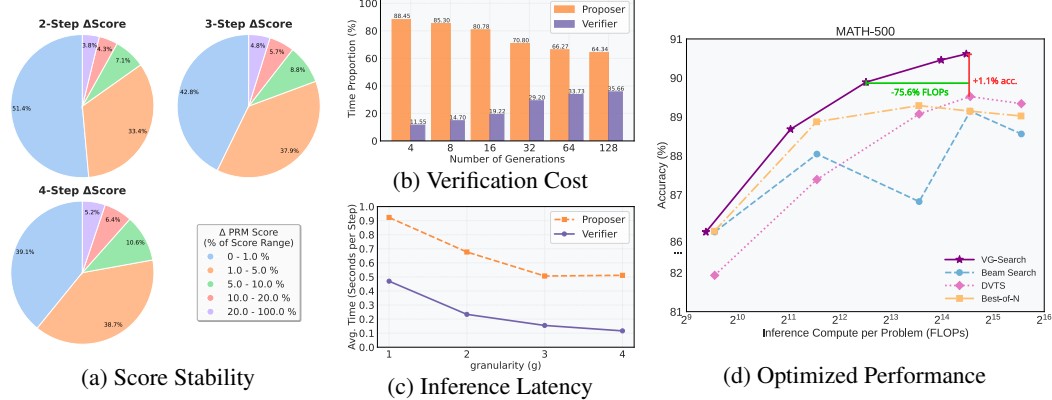

Figure 1: **Motivations for Optimizing Verification Granularity** $g$. (a) Small PRM score deltas (LLaMA3.1-8B-PRM) across generation steps indicate redundant verification. (b) Verification incurs notable latency overheads. (c) Increasing verification granularity lowers the latency of both Proposer and Verifier. (d) Optimizing $g$ improves the accuracy-compute tradeoff over fixed-granularity baselines (Qwen2.5-Math-7B as generator, Skywork-o1-1.5B as verifier).

shown in Figure 1a, our profiling results show that verifier scores often remain stable across multiple generation steps (e.g., over 50% of 2-step score differences are below $1\%$ of score range), suggesting redundancy in the current verification granularity. This inefficiency causes verification to account for an increasingly large proportion of the overall inference latency, as illustrated in Figure 1b. These insights and observations motivate us to explore the following two research questions (RQs):

- **RQ1.** Is the conventional verification granularity optimal for the accuracy–compute scaling?

- **RQ2.** If not, how can we optimize it to achieve a better accuracy–compute frontier?

To explore **RQ1**, we introduce Variable Granularity Search (*VG-Search*), a unified algorithm that generalizes verifier-guided Beam Search and Best-of-N methods. *VG-Search* employs a granularity parameter $g$ that controls the verification granularity (Section 2). Our hardware performance profiling (Figure 1c) shows that $g$ has a significant impact on the latency contributions of both the generator LLM and the verifier. To further investigate the accuracy–compute trade-off under varying $g$, we conduct extensive experiments with *VG-Search* across different values of $g$, models, compute budget, and datasets (Section 4). Building on top of the insights from **RQ1**, we propose an adaptive *VG-Search* approach which dynamically adjusts the verification granularity based on the compute budget and task attributes (Section 5) to further address **RQ2**. As shown in Figure 1d, our results demonstrate better performance–compute frontiers than previous sampling-based TTS methods. Overall, our main contributions are:

- We propose Variable Granularity Search (*VG-Search*), enabling systematic analysis of how verification granularity affects performance and compute scaling.

- Through extensive experiments, we show that the conventional static verification granularity is sub-optimal, limiting the performance scaling of verifier-guided search methods.

- We develop adaptive *VG-Search* strategies that dynamically tune granularity, achieving up to **3.1%** higher accuracy than Beam Search, **3.6%** over Best-of-N, with over **52%** compute reduction.

## 2    Exploring Verification Granularity for Reasoning

### 2.1    Verification Granularity in Verifier-Guided Search

In verifier-guided search, a generator model proposes candidate steps and a verifier model evaluates them to guide the search process. The task of generating a correct solution can be framed as a collaboration between two components:

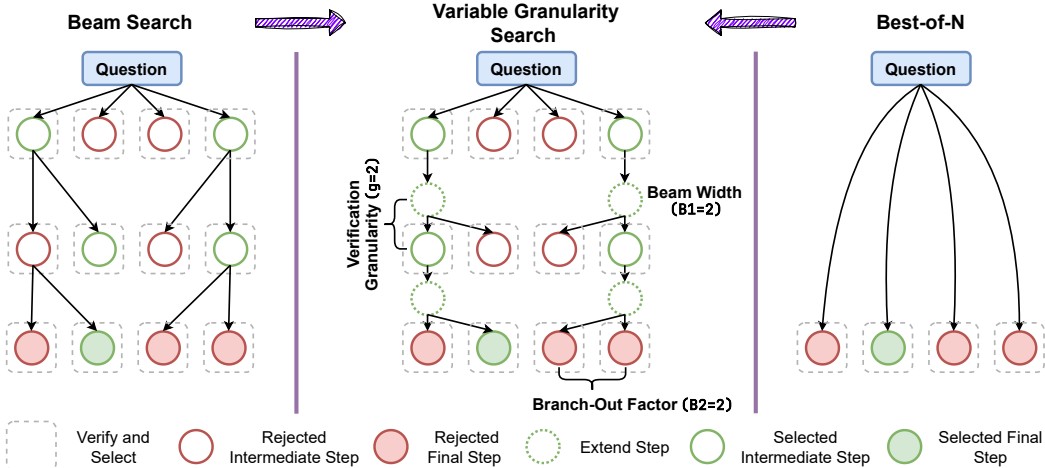

Figure 2: **Variable-Granularity Search (*VG-Search*).** *VG-Search* unifies verifier-guided Beam Search and Best-of-N via the granularity parameter $g$. At each step, $B_1 \times B_2$ candidates are verified and $B_1 = 2$ candidates are selected. $B_2 = 2$ continuations are generated per selected path. Only the $B_1$ selected candidates proceed to the **Extend Step**, reducing proposer FLOPs by pruning early.

- **Generator (Proposer)** $\mathcal{G}$: A probabilistic model, typically an LLM, responsible for generating a partial or complete candidate sequence $\mathbf{s}_{t:t+g}$, where $g$ denotes the number of the generation steps. The generation process is governed by the distribution $P(\mathbf{s}_{t:t+g} \mid \mathcal{G})$.

- **Verifier** $\mathcal{V}$: A scoring function $\mathcal{V}(\mathbf{s}_{t:t+g}) \rightarrow \mathbb{R}$ that assigns a scalar score to each partial or complete candidate sequence $\mathbf{s}_{t:t+g}$. The verifier is invoked after every $g$ generation steps. This score is intended to correlate with the expected quality or correctness of the sequence prefix.

The goal of the search process is to generate a correct or high-quality solution $\mathbf{s}_{1:T}^*$ of length $T$ via interaction between the generator and verifier. A key hyperparameter in this process is the verification granularity $g$, which controls how frequently the verifier can intervene and guide the generator. To illustrate the impact of $g$, we link the verifier-guided search to an edge case, the Infinite Monkey Theorem (IMT) [38], where a random generator (monkey) tries to reproduce a target sequence (e.g., Shakespeare's work) under a perfect verifier invoked every $g$ characters. When verification is infrequent (large $g$), the expected number of attempts grows exponentially as $\mathbb{E}[A^g]$, making success computationally infeasible. In contrast, with frequent verification, early error pruning reduces the expected cost $\mathbb{E}[A^g]$, at the expense of higher verification overhead. This toy setting underscores a fundamental trade-off between verification cost and exploration efficiency. Optimizing $g$ is thus critical for effective collaboration between generator $\mathcal{G}$ and verifier $\mathcal{V}$ under compute constraints.

The most common approach to defining $g$ relies on heuristics and remains static. However, this convention might be neither consistent nor optimal: the model distribution varies widely—some generators are more verbose, others more structured—and the number of tokens required to conceptualize a coherent "thinking step" can differ significantly across tasks. This motivates a systematic study of how performance scales with different granularity $g$.

## 2.2 Variable-Granularity Search

Beam Search [5] and Best-of-N sampling [32] are two prevalent methods for scaling large language model (LLM) performance at test-time by leveraging additional computation. These methods represent two opposing extremes within the spectrum of verifier-guided search, primarily differing in the frequency with which the verifier $\mathcal{V}$ checks the outputs from the proposer $\mathcal{G}$. In Beam Search, verification is highly fine-grained: $\mathcal{V}$ actively guides candidate selection at each generation step, enabling the early pruning of less promising paths. Conversely, Best-of-N sampling employs a coarse-grained approach, where the verification is performed on an entire candidate solution. While this allows for broader exploration, it lacks the intermediate guidance provided by frequent verification.

To bridge this gap and enable a continuous exploration of verification granularity, we introduce **Variable-Granularity Search (*VG-Search*)**. *VG-Search* unifies Beam Search and Best-of-N sam-

pling through the concept of *verification granularity* ($g$). By varying $g$, *VG-Search* allows for continuous control over the trade-off between fine-grained guidance and broad exploration. Key hyperparameters of *VG-Search* include:

- **Beam Width** ($B_1$): The number of top-scoring candidate sequences (beams) retained after verification and selection.
- **Branch-Out Factor** ($B_2$): The number of alternative continuations generated by $\mathcal{G}$ from each of the $B_1$ retained beams before the next verification phase.
- **Verification Granularity** ($g$): The number of generation steps evaluated per verification—i.e., the interval between verifier ($\mathcal{V}$) calls.

As illustrated in Figure 2, a single cycle of *VG-Search* advances the search from a candidate length $t$ to $t + g$. The full algorithm proceeds as follows:

1. **Start**: Initialize with $B_1 \times B_2$ candidates from the initial prompt.
2. **Verify & Select**: Evaluate $B_1 \times B_2$ candidates using $\mathcal{V}$, and retain the top $B_1$ beams.
3. **Extend**: For each of the $B_1$ selected beams, produce $g - 1$ generation steps using $\mathcal{G}$.
4. **Branch**: For each extended beam, produce $B_2$ single-generation-step continuations using $\mathcal{G}$.
5. **Repeat**: Go back to Step 2 and iterate until termination criteria are met.

Omitting Step 3 (**Extend**) or setting $g = 1$ reduces *VG-Search* to standard Beam Search. At the other extreme, when $g = L$ (the number of generation steps in a full solution), *VG-Search* becomes equivalent to Best-of-N sampling. One alternative design is to produce $g - 1$ generation steps in **Branch**, mimicking Beam Search but with variable verification granularity. However, prior work [32] shows that rolling out multiple future tokens, as in LookAhead Search, does not yield the best performance. We therefore adopt a simpler, more compute-efficient design: **Branch** generates just one step per candidate, and early pruning via the verifier occurs before **Extend**. As a result, only the top $B_1$ candidates continue in **Extend**, instead of the full $B_1 \times B_2$ set used in Beam Search or Best-of-N—leading to significantly lower proposer FLOPs. Larger $g$ further amplifies savings for both verifier and proposer. We validate this design in Section 4.1. A case study of *VG-Search* is illustrated in Appendix A.8.

## 2.3 Compute Cost Model for Generator and Verifier

To analyze and optimize $g$, we define a compute cost model in FLOPs with the following parameters: (1) $L$: average solution length in generation steps, (2) $P_g$, $P_v$: parameter counts of the generator and verifier, (3) $F_g$: FLOPs per generation step, (4) $F_v$: FLOPs per verifier call. The number of *VG-Search* cycles is approximately $N_{\text{cycles}} = L/g$. Following prior work [31], we approximate $F_g \approx 2TP_g$, where $T$ is the average number of tokens per generation step. Similarly, $F_v \approx 2\alpha P_v$, where $\alpha$ is a verifier-specific variable.

The total compute for generation, $C_{\mathcal{G}}$, includes extending the $B_1$ primary beams and generating $B_2$ branches across all cycles:

$$C_{\mathcal{G}} = B_1 \cdot \left( \underbrace{g - 1}_{\text{Extending}} + \underbrace{B_2}_{\text{Branching}} \right) \cdot N_{\text{cycles}} \cdot F_g = 2 \cdot T \cdot B_1 \cdot (g - 1 + B_2) \cdot \frac{L}{g} \cdot P_g \qquad (1)$$

The total compute for verification, $C_{\mathcal{V}}$, accounts for scoring $B_1 \cdot B_2$ candidates across all cycles:

$$C_{\mathcal{V}} = B_1 \cdot B_2 \cdot N_{\text{cycles}} \cdot F_v = 2 \cdot \alpha \cdot B_1 \cdot B_2 \cdot \frac{L}{g} \cdot P_v \qquad (2)$$

The total compute cost $C_{\text{total}}$ becomes:

$$C_{\text{total}} = C_{\mathcal{G}} + C_{\mathcal{V}} = 2 \cdot B_1 \cdot \frac{L}{g} \cdot [T \cdot (g - 1 + B_2) \cdot P_g + \alpha \cdot B_2 \cdot P_v] \qquad (3)$$

We adopt the proposed cost model to study the scaling behavior of *VG-Search* (Section 4), which provides a reliable proxy for measured inference latency as demonstrated in Figure 1c.

# 3 Experiment Setup

To investigate the optimality of the conventional verification granularity mentioned in *RQ1*, we conduct experiments on mathematical reasoning benchmarks.

**Task and Datasets.** We evaluate on the MATH-500 [21] and AIME [2] datasets. MATH-500 samples 500 problems from the MATH [16] benchmark, with difficulty levels labelled. AIME comprises 90 advanced high school mathematics problems from the past three years (AIME24, AIME23, AIME22). Additionally, we sample 250 problems from the MATH [16] benchmark as a validation set for Section 5, referred to as MATH-250. It consists of 50 problems per difficulty level.

**Policy and Reward Models.** For *Proposer*, we use both general-purpose model (Llama-3.2-3B-Instruct [15]) and models with internal scaling ability, fine-tuned to generate Chain-of-Thought (CoT) for math tasks, Qwen2.5-Math-7B and Qwen2.5-Math-1.5B [40]. All models are prompted to generate step-by-step solutions. For the Llama models, we use the same prompt template as the official evaluation, which explicitly states the response template. For the Qwen models, it is suggested to use a simple prompt template. Following the community convention and prior work [22, 32, 4], we use \n\n to delimit each *generation step*, which serves as the smallest unit of verification granularity. For *Verifier*, we employ discriminative Process Reward Models (PRMs), including Skywork-o1-1.5B[1] and Skywork-o1-7B[2] [27], as verifiers. Following [4], we use *Last* (final) scores for selection. Details of FLOPs calculation are in Appendix A.2.

# 4 Understanding the Limits of the Verification Granularity Convention

## 4.1 Test-Time Scaling Law with Verification Granularity

In this section, we aim to answer *RQ1*, investigating how the optimal verification granularity $g$ varies with both the compute budget and the capabilities of the *Proposer-Verifier* pair. To explore different operating points on the accuracy-compute trade-off curve (Figure 3), we vary $g$ and adjust the number of generations $n$ to modulate the total compute budget. We analyze three representative generator ($\mathcal{G}$)–verifier ($\mathcal{V}$) pairs, specifically chosen to probe the interplay between model capability and compute allocation:

1. Strong $\mathcal{G}$ (Qwen2.5-Math-7B) with a Small $\mathcal{V}$ (Skywork-o1-1.5B),

2. Weak $\mathcal{G}$ (Llama-3.2-3B-Instruct, not math-finetuned) with a Small $\mathcal{V}$ (Skywork-o1-1.5B),

3. Medium $\mathcal{G}$ (Qwen2.5-Math-1.5B) with a Large $\mathcal{V}$ (Skywork-o1-7B).

Our findings in Figure 3 indicate that the optimal verification granularity $g$ depends on both the generator's capability and the available compute budget:

**Strong Generators Prefer Sparse Verification**, while weak ones need frequent checks. For the strong Qwen2.5-Math-7B generator paired with a small verifier, while standard Beam Search ($g = 1$) is effective at lower compute costs, sparser verification ($g \in \{2, 3, 4\}$) achieves superior accuracy at medium to high compute budgets, as shown in the first column of Figure 3. Notably, $g = 3$ reaches the highest peak accuracy, outperforming $g = 1$ by approximately 4% on MATH-500. This suggests that a strong generator can reliably produce longer correct partial solutions, making frequent verification less critical and allowing compute to be reallocated (e.g., to wider beams via $B_1$) for better overall performance. As shown in the second column of Figure 3, moderately strong Qwen2.5-Math-1.5B, when paired with a large verifier, also benefits from sparser verification, with $g = 2$ consistently outperforming $g = 1$ from mid-to-high compute and achieving the best peak accuracy. However, excessively sparse verification ($g = 4$) becomes detrimental, indicating an optimal balance point. In contrast, as shown in the last column of Figure 3, the non-specialized Llama-3B generator has peak performance with frequent verification ($g = 1$) on MATH-500 and MATH-250, implying that weaker or less task-aligned generators require continuous guidance. Interestingly, on the more challenging AIME dataset, all *Proposer-Verifier* pairs show a tendency for sparser verification ($g > 1$, sometimes even $g = 4$) to outperform $g = 1$ at medium and high compute budgets. This might be because AIME problems benefit more from the broader exploration encouraged by sparser verification (which

---

[1]Skywork/Skywork-o1-Open-PRM-Qwen-2.5-1.5B

[2]Skywork/Skywork-o1-Open-PRM-Qwen-2.5-7B

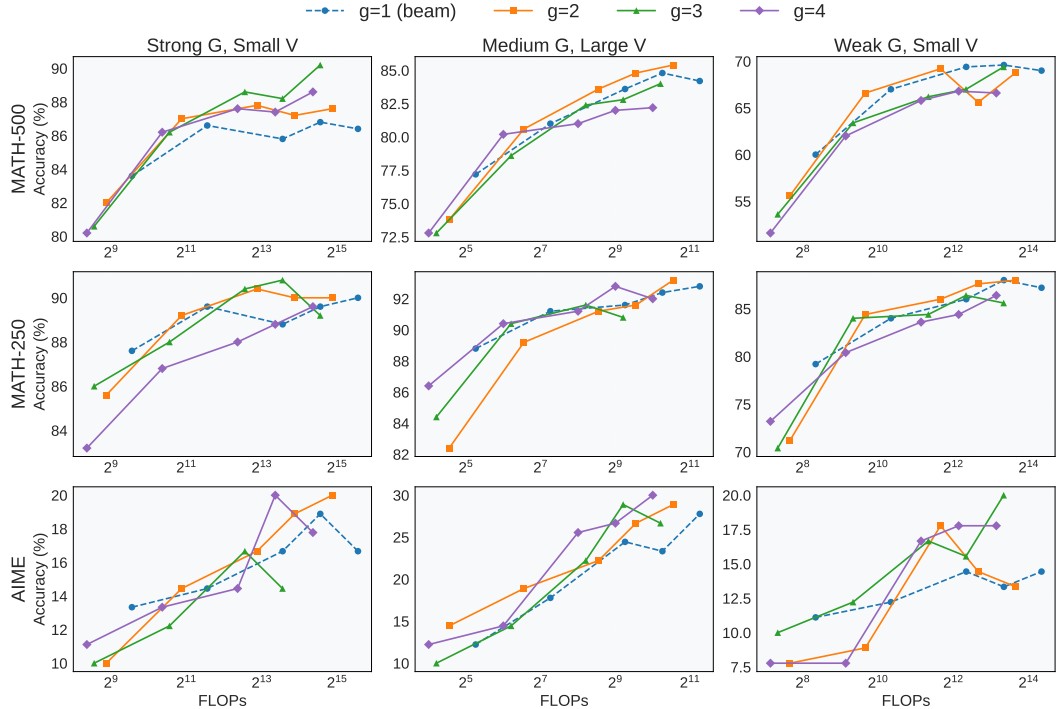

Figure 3: Accuracy vs. compute (FLOPs) across different verification granularities $g$.

makes the search resemble Best-of-N), or the PRM's step-wise utility is less consistently high on AIME, reducing the benefit of frequent checks.

**Optimal Granularity Varies with Compute Budget.** Across most models and datasets, sparser verification ($g > 1$) tends to become more competitive as the total compute budget increases. At very low compute budgets, the aggressive pruning of standard Beam Search ($g = 1$) often provides a more robust performance baseline. This aligns with the intuition that investing in more extended generation phases between verifications is more viable when more overall compute is available to support wider exploration or more candidate paths.

**Optimal Granularity Saves Computation Significantly.** A key advantage of using sparser verification ($g > 1$) is the potential for substantial FLOPs savings while maintaining—or even improving—performance. For example, in the Strong $\mathcal{G}$, Small $\mathcal{V}$ setting on MATH-500, setting $g = 3$ achieves approximately 88% accuracy at $\sim 2^{13}$ FLOPs, whereas $g = 1$ requires $\sim 2^{15}$ FLOPs to reach a slightly lower accuracy of around 87.5%. Similarly, on AIME, $g = 4$ attains comparable peak accuracy to $g = 1$, but with noticeably lower FLOPs cost across several configurations. This efficiency gain arises from fewer verifier invocations and reduced branching operations overall, as detailed in our cost model (Section 2.3). Moreover, the reduction in FLOPs effectively translates into decreased latency, as shown in Figure 1c.

**Rethinking the Optimal "Thinking Step".** The observation that increasing $g$ (sparser verification) can improve accuracy while reducing compute budget is significant. This challenges the conventional verification granularity ($g = 1$), where verification typically occurs at arbitrary delimiter boundaries (e.g., newlines), which may not always align with meaningful reasoning steps. First, as shown in Figure 1a, consecutive PRM score differences are often negligible, indicating that the current definition of a "thinking step" is overly fine-grained. Second, a larger $g$ can be interpreted as defining a more substantial and semantically coherent "thinking step". Delaying verification and branching until the end of these extended segments may avoid injecting noise by evaluating incomplete reasoning fragments, thus enabling more effective search. Thus, in response to **RQ1**, we conclude that the current convention for verification granularity is sub-optimal, motivating a more flexible definition of verification granularity—beyond fixed token-level delimiters—and highlighting the need for a more sophisticated approach to optimizing $g$ (Section 5).

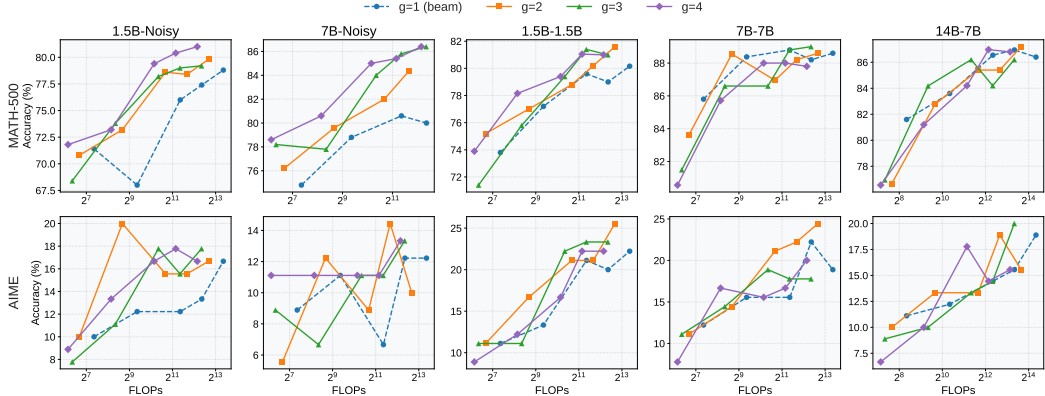

Figure 4: Ablation on verifier quality and model sizes.

## 4.2 Ablation Studies on Verifier Quality and Model Scale

We conducted two ablation studies to investigate the impact of verifier quality and model scale on the optimal granularity, with results shown in Figure 4. To simulate a worst-case scenario for verifier quality, we replaced the standard verifier with a maximally **noisy** PRM that assigned a random score to each generation step. As hypothesized, when the verifier provides no useful signal, the optimal strategy consistently shifts to **sparser verification** ($g > 1$), as frequent but unreliable guidance degrades search performance (Figure 4).

We validated our findings at different model scales by pairing various generators from the Qwen-2.5 series with Skywork-PRM verifiers of different sizes. The results (Figure 4) confirm that the optimal verification granularity remains dynamic across all tested scales.

## 4.3 Trade-offs: Granularity, Verifier Parameter, and Branching

This section further explores how $g$ interacts with other key hyper-parameters: the verifier's model parameter and the branch-out factor $B_2$ to optimize compute allocation under fixed total compute budgets. All experiments in this section utilize Qwen2.5-Math-7B as the generator.

**Granularity vs. Verifier Parameter.** We investigate whether it is more beneficial to use a strong verifier sparsely (larger $g$, larger verifier model) or a weaker verifier frequently (smaller $g$, smaller verifier model). Figure 5a compares two configurations: (1) $g = 1$ (frequent verification) with a Small Verifier (Skywork-o1-1.5B), and (2) $g = 2$ (sparser verification) with a Large Verifier (Skywork-o1-7B). As shown in Figure 5a, at lower compute budgets, sparse verification with a larger verifier yields low accuracy. However, as the compute budget increases, employing a larger verifier more sparsely achieves higher peak accuracy. This suggests a clear preference for leveraging stronger, more discerning verifiers less frequently when sufficient computational resources are available.

**Granularity vs. Branch-Out Factor** Next, we explore the trade-off between the verification granularity $g$ and the branch-out factor $B_2$, at matched total compute FLOPs. Increasing $g$ means fewer verification calls and longer generation segments, while increasing $B_2$ means exploring more alternatives at each verification point. The verifier used in this experiment is Skywork-o1-1.5B. Figure 5b shows that configurations with a small granularity ($g = 1$, $B_2 = 2$) with low branching tend to outperform other configurations at low or mid-range compute budgets. However, all configurations converge at very high compute. Comparing the above findings, a clear pattern emerges: investing the saved compute in the verifier parameter appears to be a more effective scaling strategy than simply increasing the number of branches.

## 4.4 Optimal Verification Granularity vs. Task Difficulty and Number of Samples

To assess the practical limits of sparse verification, we define the *largest effective $g$* as the maximum verification granularity that retains at least 95% of the accuracy achieved at $g=1$. We also analyze the accuracy gain of $g=2$ over $g=1$ across varying sample budgets and task difficulty levels. Figure 5d, 5e show that the optimal $g$ is strongly influenced by both difficulty and sampling budget.

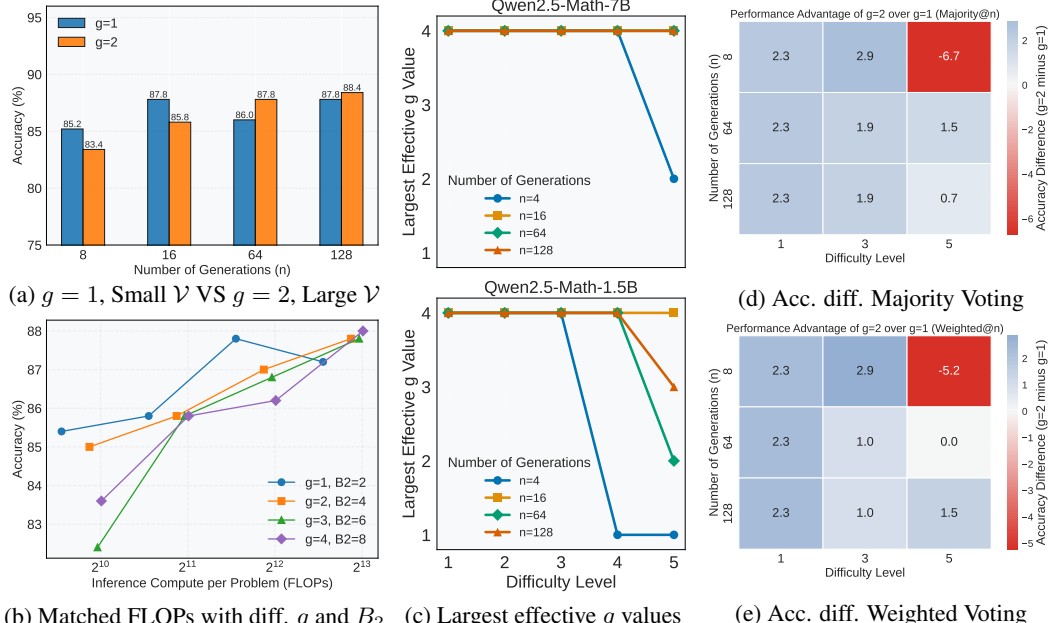

(a) $g = 1$, Small $\mathcal{V}$ VS $g = 2$, Large $\mathcal{V}$    (d) Acc. diff. Majority Voting

(b) Matched FLOPs with diff. $g$ and $B_2$   (c) Largest effective $g$ values    (e) Acc. diff. Weighted Voting

Figure 5: (a) shows $g = 2$ with a strong verifier outperforms $g = 1$ with a weak verifier at high compute budget. (b) demonstrates limited performance gain from increasing branch factor $B_2$. (c) plots the largest effective $g$ across difficulty levels for different generators. (d),(e) show the accuracy gain of sparser verification with a stronger verifier across difficulty levels and number of generations.

**Increased Task Difficulty Demands Denser Verification.** Across all models and compute budgets, we observe a consistent trend: as task difficulty increases, the largest effective $g$ decreases. For the hardest problems (Levels 4–5), $g{=}1$ or $g{=}2$ is often required to maintain performance at lower compute budget. Likewise, the advantage of $g{=}2$ over $g{=}1$ diminishes with difficulty, likely because harder tasks involve more reasoning errors, necessitating more frequent verifier intervention to stay on track.

**Higher Number of Samples Tolerates Sparser Verification.** In general, larger sample budgets allow for sparser verification, especially on easy to moderately difficult tasks. Higher budgets also amplify the performance gain of $g{=}2$ over $g{=}1$, as the generator can explore a wider search space, increasing the likelihood of finding a correct path despite less frequent verification.

## 5   Towards Adaptive Redefinition of the Verification Granularity

### 5.1   Adaptive Granularity Strategies

Our results (Figures 3, 5) show that the optimal $g$ in *VG-Search* depends on task difficulty, compute budget, and model capability. This motivates a more systematic approach to defining the search step, addressing ***RQ2***. We formulate an optimization dual for selecting verification granularity: minimizing compute for a target accuracy vs. maximizing accuracy under a compute budget. For each, we propose a corresponding adaptive granularity strategy:

**Strategy 1: Compute Minimization with Performance Parity (*CM-$g$*)** *CM-$g$* finds the largest $g^*$ that maintains accuracy within a tolerance $\epsilon$ of $g = 1$, reducing compute. Given generator, difficulty $d$, and number of generations $n$: 1. Compute baseline accuracy $\text{Acc}(g{=}1, d, n)$. 2. Increase $g$ while $\text{Acc}(g, d, n) \geq \text{Acc}(g{=}1, d, n) - \epsilon$; return the largest $g^*$ satisfying the accuracy constraint.

**Strategy 2: Accuracy Maximization with Budget Constraint (*AM-$g$*)** *AM-$g$* selects $g^*$ that maximizes accuracy under a fixed compute budget (as in number of generations): $g^* = \arg\max_{g \in \{1, \dots, g_{\max}\}} \text{Acc}(g, d, n)$.

| $n$ | Metric | Adaptive *VG-Search* | | | | Beam Search | DVTS | Best-of-N |
|---|---|---|---|---|---|---|---|---|
| | | Oracle | | Val | | | | |
| | | *CM-g* | *AM-g* | *CM-g* | *AM-g* | | | |
| 64 | Acc. | 89.2 | **90.1** | 88.6 | **90.1** | 87.0 | 89.2 | 89.5 |
| | FLOPs | **5393** | 5844 | 5543 | 6145 | 12010 | 12010 | 11952 |
| 128 | Acc. | 90.5 | **90.6** | 89.9 | 90.1 | 89.3 | 89.7 | 89.3 |
| | FLOPs | 15899 | 16200 | **11086** | 12290 | 24021 | 24021 | 23904 |
| 256 | Acc. | 90.7 | **90.8** | 90.4 | 90.4 | 88.8 | 89.5 | 89.2 |
| | FLOPs | **21572** | 22775 | 28189 | 28189 | 48042 | 48042 | 47808 |

Table 1: Performance of adaptive granularity strategies (*CM-g*, *AM-g*) vs. baselines on MATH-500, using Qwen2.5-Math-7B as the proposer and Skywork-o1-1.5B as the verifier. Best accuracy and Lowest FLOPs are **bold underlined**. $n$ is the number of generations per question.

| Budget | Strategy | 0 samples | 5 samples | 100 samples | 200 samples |
|---|---|---|---|---|---|
| $n = 64$ | CM-g | 87.03% | 88.57% | 88.57% | 88.57% |
| | AM-g | 87.03% | 88.57% | 89.19% | 90.09% |
| $n = 256$ | CM-g | 88.76% | 90.37% | 90.37% | 90.43% |
| | AM-g | 88.76% | 90.37% | 90.37% | 90.43% |

Table 2: Test accuracy on MATH-500 vs. number of validation samples used to select $g^*$. Performance converges rapidly for both medium ($n = 64$) and high ($n = 256$) compute budgets.

We compare against Beam Search, DVTS, and Best-of-N baselines on the MATH-500 test set. "val" indicates $g$ was tuned on a validation set MATH-250, while strategies under "oracle" use oracle $g$ selected on the test set. We set $\epsilon = 0$ for *CM-g*. Table 1 shows that both *AM-g* and *CM-g* improve performance and efficiency. ***AM-g* consistently achieves higher accuracy**, with gains up to 3.1% over Beam Search, while ***CM-g* provides substantial FLOPs savings**, reducing compute by over 50% at $n = 64$ and $n = 256$ while maintaining or improving accuracy. For instance, *CM-g* (val) at $n = 128$ achieves 89.9% accuracy using just 11086 FLOPs—only 46% of the baseline budget. Although oracle-tuned variants perform slightly better, validation-tuned versions (*CM-g* (val), *AM-g* (val)) still outperform fixed-$g$ methods, demonstrating strong generalization and practicality. Under our cost model, $C_{\mathcal{G}}$ dominates, so most FLOPs savings come from pruning candidate paths during the **Extend Step** (Section 2.2). In summary, adapting verification granularity $g$ to task difficulty and compute budget offers a simple yet effective way, enabling more efficient and performant LLM reasoning.

## 5.2 Practicality and Convergence of Optimal Granularity Search

A key practical consideration is the overhead of finding the optimal granularity $g^*$. We analyzed its sample efficiency by selecting $g^*$ on validation subsets of increasing size and evaluating the performance on the MATH-500 test set. The results, shown in Table 2, demonstrate that the search converges rapidly. For both high ($n = 256$) and medium ($n = 64$) compute budgets, using just 5 validation samples achieves most of the possible accuracy gains over the baseline, confirming the minimal overhead of our adaptive approach.

This search constitutes a negligible, one-time cost that is amortized over all subsequent uses, ensuring our FLOPs comparisons are fair. The practical guideline is to **reuse** a pre-computed $g^*$ for a given model, task, and budget, and **recompute** it only when a core component, like the generator model, changes. For tasks with distinct subsets (e.g., varying difficulty levels), different $g^*$ values can be pre-computed and selected at inference time, making this a practical and effective alternative to the fixed-granularity convention.

# 6 Related Work

**Test-Time Scaling (TTS).** Recent work on scaling test-time compute can be broadly categorized into two approaches [43, 19]: *i)* Internal scaling and *ii)* sampling-based TTS. **Internal scaling** methods enable models to scale their test-time compute by adapting their parameters through Supervised Fine-tuning (SFT) [42, 24] or Reinforcement Learning (RL) [33, 23, 41]. However, without explicit compute budget control [1, 3], these approaches often result in unnecessarily long CoT, even for simple queries, leading to inefficient inference costs. Sampling-based TTS, by contrast, dynamically allocates test-time computation during deployment, without requiring additional tuning to modify the model's parameters offline. **Sampling-based TTS** [10, 32, 29, 9], orthogonal to internal scaling approaches, allows for more fine-grained control over compute usage and can further improve algorithmic performance. As noted by [32, 22], multi-step lookahead approaches, such as Monte Carlo Tree Search (MCTS) [14], adaptive thought rollback [8], and hierarchical multi-step verification [36], incur substantial sampling overhead during inference, we do not consider such methods in this work.

**Verification-Guided Search.** Verifying the quality of generated outputs to guide sampling-based TTS has been shown to significantly improve the efficiency of test-time computation [22, 32, 4]. Based on the granularity of verification, existing approaches can be broadly classified into two categories: *(i)* outcome verification and *(ii)* process verification. **Outcome verification** [10] evaluates the entire response after the generation of a complete sample. This approach is widely used in sampling-based TTS, such as Best-of-N strategy [9], where a verifier selects the most promising answer from a set of full samples. While simple and effective, outcome verification provides no intermediate feedback, which can lead to wasted computation on low-quality completions. **Process verification** [21, 44], often implemented as a process reward model (PRM), evaluates intermediate reasoning steps during generation. These models assign scores to partial outputs, allowing the inference process to be guided. PRMs have been shown to improve sample efficiency and enable early pruning of unpromising token paths, leading to more compute-efficient search. Despite their promise, both outcome and process verification methods treat the verifier as a fixed component and typically apply it at a pre-defined granularity. In contrast, our work introduces an *adaptive approach that dynamically optimizes verification granularity* to maximize performance. Due to the verification inefficiency of Generative PRMs [44] under matched compute budgets [31], we leave their investigation to future work.

**Compute Optimal Scaling.** Previous research [32] indicates that the optimal choice between Best-of-N and Beam Search often depends on factors such as the available compute budget and the problem characteristics. *Yet, restricting selection to these two discrete options limits flexibility and expressiveness* in identifying an optimal compute scaling strategy. By introducing variable granularity ($g$), *VG-Search* spans a continuous spectrum of search behaviors between these extremes, offering potentially greater expressiveness for the optimal TTS strategy.

Prior work has investigated how test-time strategies, compute budgets, and problem difficulty interact, showing that optimal allocation of inference-time compute can be more effective than merely scaling model parameters [32]. Scaling the number of verification attempts per candidate has been shown to improve verification accuracy and overall task performance [30]. Recent work [31] also explores compute-optimal strategies for deciding when to generate new solutions versus verifying existing ones, taking into account the number of verification and output samples. [20] demonstrates that employing multiple diverse verifiers can enhance performance, as different verifiers are likely to detect different types of errors. [7] proposes a consistency-based model switching method that leverages multiple generator models to efficiently scale test-time compute, showing that a diverse generator pool can more effectively explore the solution space. [8] introduces adaptive thought rollback, but does not consider verification granularity. Despite the progress, optimally adapting the verifier's invocation frequency during test-time scaling remains a key unanswered question.

# 7 Conclusion

This paper investigates the critical role of the verification granularity $g$ in verifier-guided test-time scaling, introducing *VG-Search* to facilitate exploration. Our experiments show that the optimal verification granularity is highly dynamic—depending on generator capability, task difficulty, and compute budget. Building on these insights, we challenge the current convention and propose adaptive strategies for selecting $g$ to optimize efficiency and performance. This work motivates future research that explores more advanced approaches in optimizing the verification granularity $g$.

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

# A Appendix

## A.1 Optimal Compute Allocation between Proposer and Verifier

Optimizing the allocation of computational resources between proposers and verifiers is crucial for maximizing overall system performance. While previous work [31] under a fixed compute budget has primarily focused on varying the ratio of samples generated by the proposer to those evaluated by the verifier, the impact of asymmetric model capabilities—specifically, how distributing model parameters differently between the proposer and verifier affects performance—has remained largely unexplored. This section addresses this gap by investigating the optimal allocation of model parameters between these components, while adhering to a constant total compute budget, to determine how performance scales with their relative complexities.

We evaluate two configurations: Small Proposer, Large Verifier (S-P, L-V) and Large Proposer, Small Verifier (L-P, S-V), using DVTS and Beam Search. Performance is measured by accuracy against the number of samples ($n$) on tasks of varying difficulty (Level 1, Level 5) and averaged (Figure 6).

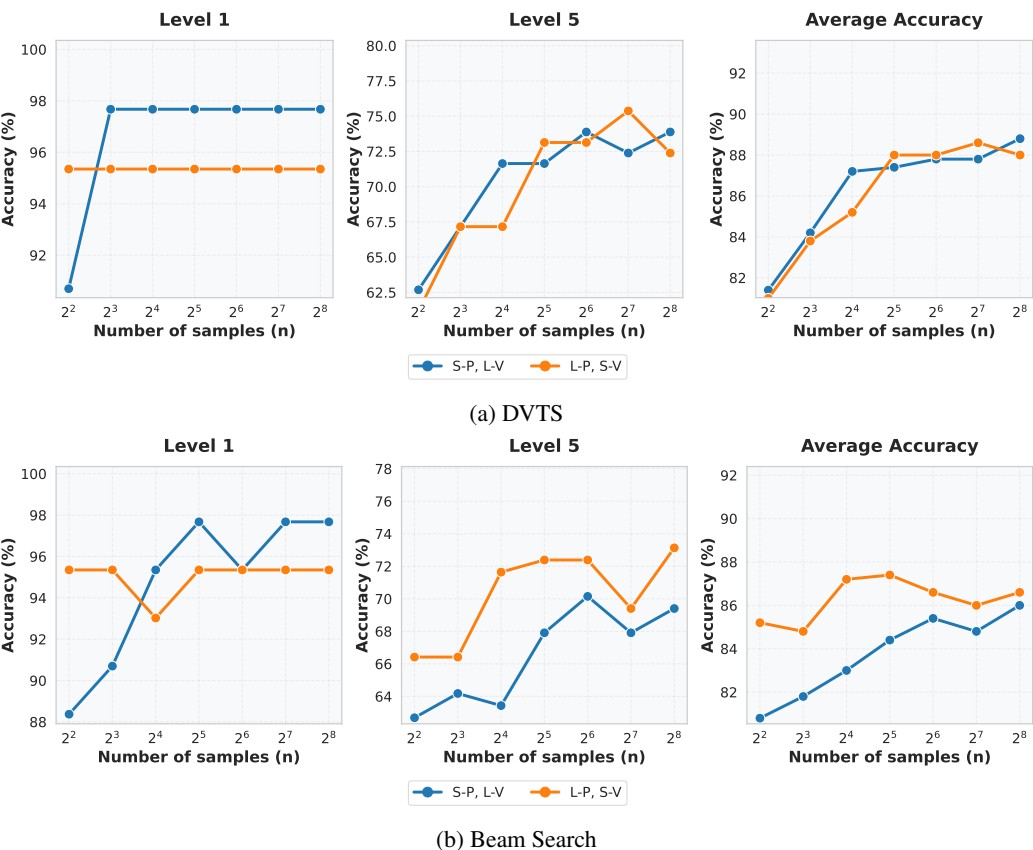

Figure 6: Comparison of Proposer-Verifier configurations under (a) DVTS and (b) Beam Search strategies. Accuracy (%) is plotted against the number of samples ($n$) for Level 1, Level 5, and Average Accuracy. Blue line: Small Proposer, Large Verifier (S-P, L-V). Orange line: Large Proposer, Small Verifier (L-P, S-V).

Key observations include:

**Verifier Quality Dictates Scaling:** With both DVTS and beam search (Figure 6), L-P, S-V starts strong on easier tasks but scales poorly with more samples ($n$), especially on harder tasks, suggesting that more samples without effective verification is counterproductive.. S-P, L-V shows more robust scaling, indicating a strong verifier better utilizes increased compute for broader exploration. A weak verifier bottlenecks the scaling.

**Strategy Determines Optimal Configuration:** The best Proposer-Verifier pairing is strategy-dependent. For DVTS, S-P, L-V generally achieves higher peak performance or scales better for complex tasks. For Beam Search (Figure 6b), L-P, S-V consistently and significantly outperforms. DVTS's diverse exploration of the solution space benefits more from a strong verifier, making a simpler proposer adequate.

**Task Difficulty Modulates Preference:** On easier tasks (Level 1), S-P, L-V can be competitive, probably because the proposer's performance is not a bottleneck for easier tasks. However, for harder tasks (Level 5), a strong proposer becomes more critical for beam search.

These results highlight that optimal compute allocation between proposer and verifier is not fixed but depends on the search strategy, task complexity, and available budget. The observation motivates more advanced compute allocation between verifiers and proposers.

## A.2 FLOPs Calculation using the Cost Model

Following prior work [31], for discriminative PRMs, we assume that $C_\mathcal{V} = 2P_v$, since the verifier outputs a single score per evaluation. Under this assumption, Equation 3 simplifies to:

$$C_{\text{total}} = 2 \cdot B_1 \cdot \frac{L}{g} \cdot \left[ T \cdot (g - 1 + B_2) \cdot P_g + B_2 \cdot P_v \right], \tag{4}$$

where $L$ is the average solution length in generation steps, $T$ is the average number of tokens per generation step. To simplify compute tracking across experiments, we define a normalized compute proxy:

$$C(g, B_1, B_2) = \frac{\lambda \cdot B_1 \cdot (g - 1 + B_2) + B_1 \cdot B_2}{g}, \tag{5}$$

where $\lambda = \frac{T \cdot P_g}{P_v}$ captures the relative cost of generation versus verification. This proxy is proportional to the actual total compute budget and is used throughout all experiments in the paper.

## A.3 Experiment Details for *VG-Search*

**Experiment Setup.** To investigate the optimal $g$ for a given difficulty level and computational budget, we conduct experiments assessing both the oracle and the validation-tuned accuracies for the *CM-g* and *AM-g* strategies. We use Qwen2.5-Math-7B [40] as proposer, and Skywork-o1-1.5B [27] as verifier. The number of generations is set to $n \in \{4, 16, 64, 128, 256\}$, with a Branch-Out Factor $B_2 = 4$, temperature 0.8, and Top-p 1.0. For each generation step, the maximum token number is set to $2,048$, employing *Last* scoring and majority voting. The number of iterations $I$ corresponds to $g$: for $g \in \{1, 2, 3, 4\}$, we set $I \in \{12, 6, 4, 3\}$ to ensure equal compute budget. The prompt template from Qwen [40] is adopted. DVTS, Best-of-N (BoN), and Beam Search are used as baselines, configured with the same parameters as *VG-Search*.

> Prompt Template for Qwen:
> {question}
> Please reason step by step, and put your final answer within \boxed{}

For ideal accuracy, we search for optimal $g$ in test dataset for every combination of number of generations and difficulty level, and compute two accuracies for each number of generation: unweighted and weighted, the former corresponds to the dataset having equal number of questions in different difficulty levels, and the latter corresponds to actual difficulty distribution in the dataset. For the searched accuracy, the only difference is that we search $g$ in the validation dataset, and obtain the accuracy in the test dataset for searched $g$.

**Baselines and Voting Methods.** Our codebase is based on HuggingFace's search-and-learn [4]. We adopt three mainstream TTS strategies: Best-of-N [5], Beam Search [32], and DVTS [4]. For all strategies we explore the number of samples of 4, 16, 64, 128, and 256. For Beam Search and DVTS, we use a fixed Branch-Out Factor $B_2 = 4$ following [32]. For all strategies, we use majority voting [37]. Temperature is set to 0.8 for all problems following HuggingFace's setup.

**Hardware and Frameworks.** We conduct our experiments on NVIDIA H100 [12] and A100 [11] GPUs, using CUDA 12.8 on Ubuntu 22.04. The vLLM library (v0.6.3) [18] is employed for model execution. The experiments for *VG-Search* on MATH-500 take from 1 to 5 hours, depending on the candidate number. To obtain accurate system-level runtime measurements, we use Nsight Systems [26] and employ the NVIDIA NVTX [13] extension to categorize execution time across different models. For the latency measurement in Figure 1c, we use Qwen2.5-Math-1.5B as generator, while using RLHFlow/Llama3.1-8B-PRM-Deepseek-Data as the verifier.

## A.4  Evaluation with an Ensemble Verifier

To broaden the scope of our evaluation, we conducted additional experiments using a multi-verifier ensemble. This approach can enhance verification quality by aggregating signals from diverse models [20]. For these experiments, we used **Qwen-2.5-Math-1.5B** as the generator. The verifier was an ensemble of two different Process Reward Models (PRMs): **Skywork-1.5B** [27] and **RLHFlow's Llama3.1-8B-PRM** [39]. The final verification score for each reasoning step was computed by averaging the scores from these two models.

The results, presented in Table 3, align with and reinforce the main conclusion of our paper. We observe that the optimal verification granularity ($g$) is dynamic and highly dependent on the compute budget (i.e., the number of generations, $n$). While dense verification ($g = 1$) is sometimes preferable at low sample counts, sparser verification ($g > 1$) consistently achieves the highest accuracy on both the MATH-500 and AIME datasets as the compute budget increases. This confirms that our findings on adaptive verification granularity generalize even when using a stronger, ensembled verifier.

| n | MATH-500 (%) | | | | AIME (%) | | | |
|---|---|---|---|---|---|---|---|---|
| | g=1 | g=2 | g=3 | g=4 | g=1 | g=2 | g=3 | g=4 |
| 4 | **76.0** | 73.2 | 71.6 | 69.6 | 12.22 | 8.89 | 11.11 | **13.33** |
| 16 | **80.0** | 77.4 | 76.4 | 78.4 | 13.33 | 13.33 | **16.67** | 14.44 |
| 64 | 79.4 | 80.4 | 80.8 | **81.0** | 16.67 | 16.67 | 16.67 | **18.89** |
| 128 | 80.8 | **82.2** | **82.2** | 80.8 | 15.56 | **24.44** | 23.33 | 18.89 |
| 256 | 80.2 | 80.0 | 80.2 | **81.2** | 23.33 | 23.33 | 23.33 | **23.33** |

Table 3: Accuracy of VG-Search with an ensemble verifier across MATH-500 and AIME datasets. For each number of generations ($n$), the highest accuracy in each row is **bolded**, indicating the performance achieved at the optimal granularity $g$.

## A.5  Applicability to Tasks with Outcome-Level Feedback

VG-Search is applicable to tasks with outcome-level feedback, provided a PRM can be trained on intermediate generation steps. In such scenarios, the core search algorithm remains unchanged. However, for open-ended generation tasks like code synthesis where outputs are diverse, standard majority voting is not feasible for final answer selection. A practical alternative is to select the single candidate trajectory with the highest cumulative or final PRM score.

To demonstrate this, we evaluated VG-Search on the **HumanEval** benchmark using **LLaMA-3.2-1B** as the generator and **Skywork-o1-1.5B-PRM** as the verifier. We used newline characters to delimit generation steps. The results, reported as pass@1 scores in Table 4, show that sparser verification ($g > 1$) can outperform dense verification ($g = 1$), particularly as the compute budget ($n$) increases. This confirms that our central conclusion—that the optimal verification granularity is dynamic—generalizes to open-ended generation tasks.

## A.6  Standard Deviation with Multiple Trials

To evaluate the robustness of our results, we conducted experiments over multiple trials using different random seeds. This analysis provides a more rigorous assessment of variance and confirms the stability of our conclusions. The following results were generated using the **Qwen-Math-7B** generator and the **Skywork-o1-1.5B PRM** on the MATH-500 dataset, averaged over three distinct runs.

| n | g=1 | g=2 | g=3 | g=4 |
|---|---|---|---|---|
| 4 | 0.250 | **0.274** | 0.238 | 0.220 |
| 16 | **0.335** | 0.305 | 0.268 | 0.311 |
| 64 | 0.256 | **0.335** | 0.274 | 0.305 |

Table 4: Pass@1 scores on HumanEval using VG-Search, selecting the highest-scoring candidate. The best performance for each compute budget ($n$) is **bolded**.

Table 5 shows the mean accuracy and standard deviation for this experiment. The results from multiple trials are consistent with our original findings, demonstrating the stability of the observed trends. The low standard deviations across all configurations reinforce this conclusion. While dense verification ($g = 1$) performs well at smaller compute budgets, **sparser verification** ($g > 1$) consistently achieves higher accuracy as the number of samples increases ($n \geq 64$).

| n | g=1 | g=2 | g=3 | g=4 |
|---|---|---|---|---|
| 4 | **84.13 ± 0.50** | 82.27 ± 0.24 | 81.73 ± 0.39 | 80.40 ± 0.43 |
| 16 | **86.80 ± 0.28** | 86.13 ± 0.22 | 86.13 ± 0.25 | 86.40 ± 0.16 |
| 64 | 86.20 ± 0.33 | 87.27 ± 0.15 | **88.27 ± 0.34** | 87.87 ± 0.19 |
| 128 | 86.67 ± 0.19 | 87.60 ± 0.57 | 88.27 ± 0.21 | **88.47 ± 0.32** |
| 256 | 86.67 ± 0.19 | 87.93 ± 0.34 | **89.13 ± 0.27** | 88.73 ± 0.19 |

Table 5: Mean accuracy ± standard deviation over 3 runs with different seeds on MATH-500. The highest mean accuracy for each sample count ($n$) is **bolded**.

## A.7 Limitations

While our proposed adaptive strategies to optimize the verification granularity $g$ based on overall problem characteristics like difficulty and compute budget, a key limitation is that the chosen $g$ remains fixed throughout the generation process for a single problem instance. This approach does not account for potential variations in reasoning complexity or generator confidence within different stages of solving a single problem. A more sophisticated approach, potentially offering further efficiency gains, would involve dynamically adjusting the granularity $g$ on a step-by-step basis during inference, adapting verification frequency based on real-time signals. Developing such fine-grained, intra-problem adaptive granularity remains an important direction for future research.

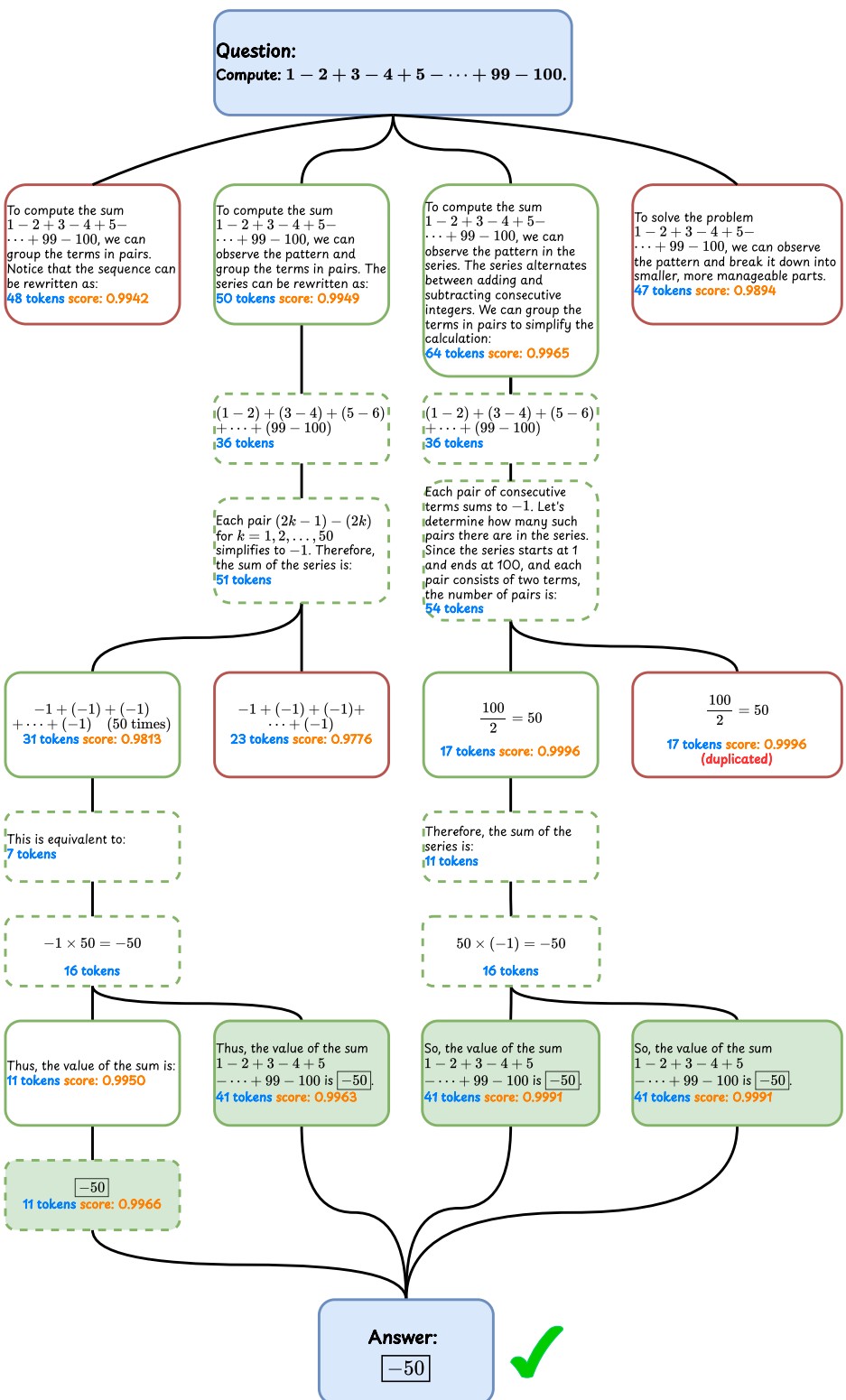

Figure 7: A case study of *VG-Search*. $B_1 = 4, B_2 = 2, g = 3$

