# OpenReview forum: "Rethinking Optimal Verification Granularity for Compute-Efficient Test-Time Scaling"
_NeurIPS.cc/2025/Conference — NeurIPS 2025 poster_

### Official Review · Reviewer_jWiE · 2025-06-25

**Clarity:** 4
**Significance:** 3
**Originality:** 3
**Rating:** 4
**Confidence:** 4

**Summary:**

The authors raise a new line of tuning during test-time compute inference for large language models: the frequency and strength of a *verifier*, which assigns a score to a reasoning step performed by an LLM (often from a pretrained reward model.) The authors demonstrate that existing methods of verification in TTC lead to a number of inefficiencies, most salient of which is a computational inefficiency where verification steps at too fine a granularity lead to wasted FLOPs with little information gain (since consecutive reasoning steps often highly correlate with correctness). To address this, the authors systematically analyze how varying verification granularities within their search technique, VG-Search, can impact computational efficiency and accuracy. The authors show that a granularity in-between every-step and once at the end often leads to the best performance in trading off compute available to the LLM itself v. the verifier. The authors also introduce adaptive VG-search, where compute or accuracy constraints can guide the search for the optimal verification granularity, and show that these found granularity values effectively trade off less compute or higher accuracy in comparison to existing (naive) methods.

**Questions:**

* Can you explain why rejected intermediate steps in beam search get expanded on in Fig. 2? I thought that rejected steps shouldn't be further expanded on by beam search.
* The usage of the verifier is not fully clear to me (as mentioned in the intro and then again on line 67.) What threshold do you use to decide whether to accept or reject a reasoning step?
* Can you provide more intuition in line 116 regarding the choice of the simpler algorithmic design: is it because some solutions just start out as incorrect and you want to catch these early?
* Line 256: How does epsilon get selected?
* See questions regarding section 5. Do the reported numbers don't report the computational overhead required during the g* search process?

**Ethical Concerns:**

["NO or VERY MINOR ethics concerns only"]

**Final Justification:**

The additional experiments and points raised by the authors to help clarify and further robustify the claims made in the paper are satisfactory to me and as such I will maintain my score.

**Limitations:**

The limitations are briefly spelled out in the appendix. One of the limitations discussed brings about the same (similar) point I made in my comments on Section 5.

**Quality:**

3

**Strengths And Weaknesses:**

Strengths:

* I find that the introduction of the new axis of verification granularity is valuable as a standalone contribution and important in test-time compute optimization. This parameter is something that the community should focus on and bring into existing TTC optimization methods to further analyze and improve performance.
* The paper is broadly quite well written and easy to follow. The contribution is straightforward, but I view this as a strength: an intuitive insight into further optimizing TTC can bring about expected and desired tradeoffs in compute and accuracy.
* The experimental results do tell a good story that backs up the claims of the authors. I found especially interesting that there was not always a consistently best granularity in Figure 3 when fixing the strength of the generator which strongly motivates the methods developed in section 5.
* Sections 3-4 (and most of 2) are very detailed and clear from a technical standpoint without becoming pedantic or unnecessarily rigorous.

Weaknesses:
* There are some details that still lead to confusion on my end that I'd like to have clarified (see questions section.)
* I felt the toy example in line 74 is not the demonstrably best choice. With infinite monkeys, there is almost certainly a need for a very fine granularity when it comes to verification since they’ll probably get the next step wrong - perhaps a different example would more strongly motivate the need for a tradeoff.
* Section 2.3 is not particularly insightful and could probably be condensed to just a few lines. it’s pretty easy to follow how a reduction of verifier calls can scale down test time compute.
* The choice of generator and verifier pairings in section 4 don't make a ton of sense to me and I'd like to see more pairings. At the very least, seeing performance by the most performant generator paired with the most performant verifier would be valuable here.
* The biggest issue I have is with section 5, which I think is much too short given its significance with respect to the rest of the paper and lacks many clarifying details. More analysis here would really strengthen the paper. Namely: how long do the AM-g and CM-g methods take to converge? What is the computational overhead of finding the optimal g* before actually deploying that on the test/validation set? I assume that the reported numbers are all after g* is actually found, but finding this g* always practical? Is it a fair comparison if these numbers don’t include that burn-in cost to find the optimal g*? And if the reported numbers don't report the computational overhead required during the g* search process, there should be a thorough discussion and analysis on when and how to use g* during actual execution. When do we reuse g* during live inference? When do we compute a new g*? Is it for each finetuned model? What if a finetuning dataset has subsets that may require different g*?

Minor:
* Lines 100-102: Variable names B1 and B2 are somewhat obscuring naming choices
* The lead takeaway in line 168 isn't the most insightful. I'd instead lead with the takeaway regarding how different granularities are optimal across different pairings/baselines and characterize why this might be the case.

Overall, I think the new line of insight presented by the authors is something that the NeurIPS and broader community should pay attention to. With that being said, I think the work can be greatly strengthened with additional clarity and analysis and my score will reflect that.

---

> ### Author Rebuttal · Authors · 2025-07-29
>
> We thank the reviewer for the helpful feedback and will clarify the points of confusion as follows.
>
> > Q1: Example of infinite monkey might not be the best choice for demonstration.
>
> We appreciate your opinion. We agree with you that the infinite monkey problem does not represent the general scenario of verifier-guided search with capable LLMs.
>
> Our intention with this example, as we briefly touch on in the paper, is to motivate the importance of verification granularity that balances verification cost and exploration cost. The "infinite monkey" case illustrates this in its most extreme form. The rest of our paper then explores how the optimal balance of this trade-off shifts away from dense verification when the generator is not random but a powerful LLM.
>
> To avoid confusion, we will revise this section to more clearly frame the example as a simplified illustration of this fundamental trade-off, explicitly stating that the optimal strategy changes for capable models.
>
> > Q2: Section 2.3 could have been shorter
>
> We appreciate your opinion as a thoughtful reader. We will minimize the length of Section 2.3 in our next version to make it more information-dense and concise.
>
> > Q3: The motivation for the choice of generator and verifier pairings isn’t clear, and more pairings are needed
>
> We agree that a more comprehensive set of pairings strengthens our claims. Our initial pairings were chosen to study the optimal granularity with comparable compute FLOPs. However, based on your feedback, we have conducted new experiments to provide a full grid analysis of generator and verifier strength.
>
> In the next version of the paper, our main results will feature a complete set of pairings:
>
> * **Qwen-Math-1.5B \+ Skywork-1.5B-PRM** *(New Experiment)*
> * **Qwen-Math-7B \+ Skywork-7B-PRM**  *(New Experiment, addressing your specific request)*
> * **Qwen-Math-1.5B** \+ **Skywork-7B-PRM**  *(Original Paper)*
> * **Qwen-Math-7B** \+ **Skywork-1.5B-PRM**  *(Original Paper)*
>
> Results on **Qwen-Math-1.5B \+ Skywork-1.5B-PRM**
>
> | n | MATH-500 |  |  |  | AIME |  |  |  |
> | :---- | :---- | :---- | :---- | :---- | :---- | :---- | :---- | :---- |
> |  | **$g$=1**  | **$g$=2**  | **$g$=3**  | **$g$=4**  | **$g$=1**  | **$g$=2**  | **$g$=3**  | **$g$=4**  |
> | 4 | 73.8 | 75.2 | 71.4 | **73.90** | 11.11% | **11.11%** | **11.11%** | 8.89% |
> | 16 | **77.2** | 77.0 | 75.8 | 78.16 | 13.33% | **16.67%** | 11.11% | 12.22% |
> | 64 | **79.6** | 78.8 | 79.4 | 79.4 | 21.11% | 21.11% | **22.22%** | 16.67% |
> | 128 | 79.0 | 80.2 | **81.4** | 81.05 | 20.00% | 21.11% | **23.33%** | 22.22% |
> | 256 | 80.16 | **81.6** | 81.0 | 81.0 | 22.22% | **25.56%** | 23.33% | 22.22% |
>
> Results on **Qwen-Math-7B \+ Skywork-7B-PRM**
>
> | n | MATH-500 |  |  |  | AIME |  |  |  |
> | :---- | :---- | :---- | :---- | :---- | :---- | :---- | :---- | :---- |
> |  | **$g$=1**  | **$g$=2**  | **$g$=3**  | **$g$=4**  | **$g$=1**  | **$g$=2**  | **$g$=3**  | **$g$=4**  |
> | 4 | **85.8** | 83.6 | 81.49 | 80.56 | **12.22%** | 11.11% | 11.11% | 7.78% |
> | 16 | 88.38 | **88.55** | 86.6 | 85.71 | 15.56% | 14.44% | 14.44% | **16.67%** |
> | 64 | **88.8** | 86.95 | 86.6 | 88.0 | 15.56% | **21.11%** | 18.89% | 15.56% |
> | 128 | 88.2 | 88.2 | **88.8** | 88.0 | **22.22%** | **22.22%** | 17.78% | 16.67% |
> | 256 | 88.6 | **88.6** | **89.0** | 87.8 | 18.89% | **24.44%** | 17.78% | 20.00% |
>
> Across all four pairings, the data consistently shows that the optimal verification granularity is not fixed but is a dynamic parameter that depends on the intricate interplay among compute budget, task, and the specific generator-verifier pair. We also include results using Qwen-2.5-14B-Instruct as the generator in the response to Q1 of Reviewer RT2j.
> This comprehensive analysis will be integrated into the main results section of our revised paper, providing a much stronger empirical foundation for our claims. Thank you for helping us improve our work.
>
> > Q4: Clarity on Section 5
>
> Thank you for emphasizing the importance of Section 5 and for your constructive questions. We agree that a deeper analysis of the adaptive strategies strengthens the paper and will clarify the settings and implementation details in the revision. Please see our responses below.
>
> > Q5: Convergence of AM-g and CM-g and computational overhead of finding optimal $g\*$
>
> To clarify how many validation samples are needed to find a good granularity $g\*$, we conducted a convergence analysis. We simulated finding $g\*$ on subsets of our validation set (MATH-250) of increasing size and then evaluated the performance on the MATH-500 test set.
>
> Our findings show that the search for an effective $g\*$ is highly sample-efficient and converges very quickly.
>
> The tables below show the test accuracy for n=256 and n=64 as a function of the number of validation samples used to select $g\*$.
>
> When **n=256**
>
> |  | 0 sample (Beam Search) | 5 samples | 100 samples | 250 samples |
> | :---- | :---- | :---- | :---- | :---- |
> | CM-g | 88.76 | 90.37 | 90.37 | 90.43 |
> | AM-g | 88.76 | 90.37 | 90.37 | 90.43 |
>
> When **n=64**
>
> |  | 0 sample (Beam Search) | 5 samples | 125 samples | 200 samples |
> | :---- | :---- | :---- | :---- | :---- |
> | CM-g | 87.03 | 88.57 | 88.57 | 88.57 |
> | AM-g | 87.03 | 88.57 | 89.19 | 90.09 |
>
> The optimal granularity g∗ converges quickly with minimal validation data, across different compute budgets:
>
> * **High compute budget (n \= 256):** Accuracy improves from 88.76% to 90.37% with just 5 validation samples. Increasing to 250 samples yields only a marginal gain (90.43%), indicating rapid convergence.
>
> * **Medium compute budget (n \= 64):** A similar pattern holds. Accuracy rises from 87.03% to 88.57% using 5 samples. While AM-g benefits slightly from more data (up to 90.09%), most of the gain comes early.
>
> The fast convergence aligns with our main conclusion that dense verification is often not optimal, especially under a high compute budget. It demonstrates that the additional compute overhead is minimal and finding $g\*$ is practical.
>
> > Q6: Fairness and overhead of finding $g\*$
>
> Our adaptive approach finds the optimal granularity $g\*$ once per task and generator-verifier pair using a very small number of validation samples (Q5). This is a one-time cost that is negligible compared to live inference and can be reused across all future test cases.
>
> This ensures that the FLOPs comparisons in Section 5 are fair, as the "burn-in cost" is insignificant in any practical scenario:
>
> * **Minimal Overhead:** Validation requires few samples, incurring negligible cost. In practice, it takes less than **5 minutes** on a single H100 for our experiments in Section 5\.
>
> * **Amortized Use:** Once found, g∗ is reused, making the cost insignificant over time.
>
> We will include these analyses and discussions in the next version of the paper as requested. Thank you for helping us improve the paper.
>
> > Q7: When to Recompute or Reuse $g\*$
>
> The general guideline for applying $g\*$ in practice is:
>
> 1) Reuse $g\*$: For a given fine-tuned model, verifier, task domain, compute-budget, and problem characteristic (in this case, problem difficulty), the same $g\*$ can be reused indefinitely.
> 2) Recompute $g\*$: A new $g\*$ should be computed only when one of the core components changes (e.g., you switch to a new generator model).
>
> For datasets with distinct subsets (e.g., varying difficulty levels), a more fine-grained strategy can be employed, depending on the characteristics of the task. As in Table 1, one can compute a different $g\*$ for each difficulty subset and switch between them at inference time based on the problem's difficulty level.
>
> We agree that more advanced methods for selecting g are a promising future direction. As we note in our limitations section and our response to Reviewer vtyM, dynamically adapting $g\*$ during a single generation based on real-time signals (like model confidence) is an exciting next step. The methods proposed in this paper represent an attempt to challenge the fixed-granularity convention and provide a simple, practical, and effective alternative.
>
> > Q8: How different granularities are optimal across different pairings/baselines
>
> We added a fully random verifier as a special case study. Please see our response to Reviewer vtyM’s Q4. We observed that for unreliable verifiers, sparser verification is preferred in most cases. We will refine the takeaways in Section 4.1 to better emphasize the trends across different experimental pairings.
>
> > Q9: The clarity of variable names
>
> Thank you for the helpful suggestions on presentation and clarity. We will revise the variable names used in our search parameters.
>
> > Q10: Why rejected intermediate steps in beam search get expanded on in Fig. 2
>
> You are absolutely correct—we do not expand rejected steps. The node in Figure.2 is the step generated for verification (before extending). We will make it clearer.
>
> > Q11: How to decide whether to accept or reject a reasoning step
>
> We keep the top K reasoning steps according to their scores from PRM. We will make it clearer in our next version.
>
> > Q12: Why do we early prune the beam? is it because some solutions just start out as incorrect and you want to catch these early?
>
> Yes, this is the motivation of our approach. We want to catch incorrect beams early, before spending more compute (extending) on them.
>
> > Q13: How epsilon is selected (line 256)
>
> The hyperparameter epsilon controls the trade-off between accuracy and compute savings. It is user-defined and depends on the task’s sensitivity to error. In Section 5, we set it to 0.
>
> > Q14: overhead required during the g* search process?
>
> Please refer to Q5, Q6, and Q7.
>
> > I think the work can be greatly strengthened with additional clarity and analysis and my score will reflect that
>
> We hope that the new experiments and clarifications provided have offered the additional analysis and clarity needed to strengthen the paper as you suggested. Please feel free to let us know if you have further questions.

---

> > ### Comment · Reviewer_jWiE · 2025-08-04
> > **Thanks for your response**
> >
> > Thanks to the authors for the additional results and clarifications. I think the additional discussion (particularly on g* and the clarification of some details regarding VG-search) will help strengthen the paper quite a bit.

---

> > > ### Author Response · Authors · 2025-08-06
> > > **Thank You & Final Questions**
> > >
> > > Thank you again for your time and insightful suggestions. We will include the expanded discussions in the final manuscript.
> > >
> > > As the discussion period approaches the end, we are happy to address any remaining questions you may have.

---

### Official Review · Reviewer_RT2j · 2025-06-30

**Clarity:** 3
**Significance:** 2
**Originality:** 2
**Rating:** 4
**Confidence:** 3

**Summary:**

This work studies LLM test-time scaling for reasoning tasks by repeated sampling and verification.
The authors argue that dense verification in standard beam search can be wasteful,
and propose a generalized variant that allows verification at variable granularity:
for each branch, generate $g \ge 1$ steps before verification and pruning.
Empirical results confirm that higher accuracy and efficiency can be achieved by selecting the granularity properly.

**Questions:**

None

**Ethical Concerns:**

["NO or VERY MINOR ethics concerns only"]

**Final Justification:**

The authors have addressed my concerns, and I would maintain my rating.

**Limitations:**

Yes, the authors discussed in Appendix A.4 that the hyperparameter g (granularity) is chosen once (e.g., by optimizing performance on a validation set) and then fixed throughout at inference time.
Finding a way of adaptively setting this hyperparameter may require further work.

**Quality:**

3

**Strengths And Weaknesses:**

**Strengths:**

* The proposed method is sensible, simple yet effective.
* The empirical results are promising.
* The overall writing is clear.
* There are some good explanations, intuitions and insights, for example:
    * The authors spot redundant verification in beam search, as shown in Figure 1;
    * Lines 168 - 184 explain why stronger generators can benefit more from sparse verification;
    * Lines 202 - 213 explains why dense verification at each step is likely sub-optimal.


**Weaknesses:**

There is room for improvement in experiments, for example:

* Empirical evaluation is done with limited model scale, up to 7B. Nonetheless, the rationale of the proposed method seems to hold regardless of model scale.
* Each experimental result is obtained with a single trial, rather than averaged over multiple independent trials. As a result, some curves are a bit messy (e.g., Fig 1(d) and Fig 3), and small differences (e.g., 1% in accuracy) might be partially due to randomness of experiments.
* A related issue is that, for Question 7 in the checklist regarding statistical significance, the authors say "Our main results do not contain randomness as we fixed the seed". This is obviously not a valid argument (as the choice of seed itself can be random); otherwise, any empirical study can refrain from running multiple trials and reporting statistical significance, by the same argument.


Nonetheless, it is understandable that computational costs can be high for additional experiments with larger model scales or larger number of trials.

---

> ### Author Rebuttal · Authors · 2025-07-29
>
> We appreciate the reviewer’s thoughtful feedback and will clarify the points of confusion as follows.
>
> > Q1: Scaling model sizes
>
> To enrich the evaluation scope with additional model scales, we conducted experiments using a larger model **Qwen-2.5-14B-Instruct**, paired with the Skywork-o1-7B PRM. Given the time constraints of the rebuttal period, we prioritized this 14B model.
>
> Our new findings, summarized below, align well with our paper's main conclusions:
> | n | Optimal g (MATH-500) | Top Accuracy | Optimal g (AIME) | Top Accuracy |
> | :---- | :---- | :---- | :---- | :---- |
> | 4 | 1 | 81.6 | 1 | 11.11 |
> | 16 | 3 | 84.2 | 2 | 13.33 |
> | 64 | 1 | 86.6 | 4 | 17.78 |
> | 128 | 4 | 87.0 | 2 | 18.89 |
> | 256 | 2 | 87.2 | 3 | 20.00 |
>
> Here is a complete table of results:
>
> | n | MATH-500 |  |  |  | AIME |  |  |  |
> | ----- | ----: | ----: | ----: | ----: | ----: | ----: | ----: | ----: |
> |  | **g \= 1** | **g \= 2** | **g \= 3** | **g \= 4** | **g \= 1** | **g \= 2** | **g \= 3** | **g \= 4** |
> | 4 | **81.6** | 76.7 | 77.0 | 76.6 | **11.11** | 10.00 | 8.89 | 6.67 |
> | 16 | 83.6 | 82.8 | **84.2** | 81.2 | 12.22 | **13.33** | 10.00 | 10.00 |
> | 64 | **86.6** | 85.4 | 86.2 | 84.2 | 14.44 | 13.33 | 13.33 | **17.78** |
> | 128 | **87.0** | 85.40 | 84.20 | **87.0** | 15.56 | **18.89** | 14.44 | 14.44 |
> | 256 | 86.4 | **87.2** | 86.2 | 86.8 | 18.89 | 15.56 | **20.00** | 15.56 |
>
> These results confirm that our main conclusion holds at a larger scale. Specifically, we observe that the optimal verification granularity varies with the number of samples, with larger sample sizes favoring sparser verification. This reinforces our central claim that the optimal granularity is dynamic and not fixed at $g$=1.
>
> We will integrate these new 14B-scale results into the main body of the paper to strengthen our empirical evaluation. If the reviewer believes that further experiments with models of specific sizes would be valuable for the final version, we would be happy to include them.
>
> > Q2: Multiple trials of experiments
>
> We thank the reviewer for the suggestion to include results from multiple trials. We agree that this provides a more robust evaluation.
>
> Following this recommendation, we have conducted new experiments with multiple seeds to analyze the variance of our results. Below are the updated accuracies (mean ± std. dev. over 3 runs with different seeds) for the Qwen-Math-7B generator with Skywork-o1-1.5B PRM on the MATH-500 dataset.
>
> | n | $g$=1 | $g$=2 | $g$=3 | $g$=4 |
> | :---- | :---- | :---- | :---- | :---- |
> | 4 | 84.13 ± 0.50 | 82.27 ± 0.24 | 81.73 ± 0.39 | 80.40 ± 0.43 |
> | 16 | 86.80 ± 0.28 | 86.13 ± 0.22 | 86.13 ± 0.25 | 86.40 ± 0.16 |
> | 64 | 86.20 ± 0.33 | 87.27 ± 0.15 | 88.27 ± 0.34 | 87.87 ± 0.19 |
> | 128 | 86.67 ± 0.19 | 87.60 ± 0.57 | 88.27 ± 0.21 | 88.47 ± 0.32 |
> | 256 | 86.67 ± 0.19 | 87.93 ± 0.34 | 89.13 ± 0.27 | 88.73 ± 0.19 |
>
> The results averaged over multiple runs confirm our original conclusions. While dense verification ($g$=1) is effective at lower compute budgets, sparser verification ($g$\>1) consistently leads to higher accuracy as the compute budget increases (for n ≥ 64). The small standard deviations across runs also highlight the stability of these trends.
>
> We will make our best effort to conduct additional multi-trial experiments in future versions to improve the robustness of our evaluation. If the reviewers would like to see multi-trial results for any specific experiments, we will do our best to include them.

---

> > ### Comment · Reviewer_RT2j · 2025-08-05
> >
> > I would like to thank the authors for addressing my concerns.
> > I'm inclined to maintain my current (positive) rating for now, although it might be updated if necessary during the AC-reviewer discussion period.

---

> > > ### Author Response · Authors · 2025-08-06
> > > **Thank You & Final Questions**
> > >
> > > Thank you for your response. Please let us know if you have any unaddressed questions. We will be happy to answer them.

---

### Official Review · Reviewer_vtyM · 2025-07-03

**Clarity:** 3
**Significance:** 3
**Originality:** 3
**Rating:** 5
**Confidence:** 3

**Summary:**

The paper makes the observation that the two most prominent Test-Time-Scaling methods, Beam Search, and Best-Of-N, can be unified into a single framework, where the “granularity” (or frequency of verification steps) is varied. Beam Search invokes the verifier at each step (granularity 1), and Best-Of-N invokes the verifier at the end of each generation. The paper very reasonably questions whether a verification frequency between the two might yield more cost effective TTS, and introduces VG-Search, which parametrizes the granularity. The main findings are that stronger generators, easier tasks, and larger sample budgets tolerate higher granularity.

**Questions:**

- Could you give more detail on how a dynamic adjustment of granularity might be implemented? For instance, could the verifier's confidence score on a partial solution be used as signal to determine he length of the next generation segment before the next verification?
- Your experiments are focused on mathematical reasoning. How do you expect your findings on optimal verification granularity to generalize to other domains, such as creative writing, code generation, or long-form question answering?
- The analysis assumes an accurate Process Reward Model (PRM). How would the best granularity change if the verifier itself is noisy or poorly calibrated?

**Ethical Concerns:**

["NO or VERY MINOR ethics concerns only"]

**Final Justification:**

Thank you to the authors for the thoughtful rebuttal and followup experiments. I will keep my score (5) of Accept.

**Limitations:**

Yes.

**Quality:**

3

**Strengths And Weaknesses:**

Strengths:
- The paper addresses a novel and significant aspect of verifier-guided test time scaling, which is the frequency of verification steps (granularity.) It frames existing methods (Beam-Search and Best-Of-N) as two extremes of a unified perspective, and introduces VG-Search to interpolate between the methods.
- VG-Search gives a practical way to improve the accuracy-compute trade-off, with up to a 3.6% accuracy gain or a 52% reduction in FLOPs.

Weaknesses:
- The "atomic unit" of generation and verification is based on newline characters, whereas it may be useful to consider more semantically meaningful notions of a "step" (this is acknowledged by the authors.)
- Beyond varying granularity by task difficulty, it would be interesting to consider varying granularity dynamically during execution of a task or within a generation process.

---

> ### Author Rebuttal · Authors · 2025-07-29
>
> We appreciate the reviewer’s thoughtful feedback and will clarify the points of confusion as follows.
>
> > Q1: Semantically relevant notion of a “step”
>
> We thank the reviewer for this insightful suggestion. We fully agree that defining a “step” based on semantic meaning is both valuable and promising—a perspective that also serves as the core motivation of this paper.
>
> For this work, we adopted the newline character delimiter for the following reasons:
>
> 1. **Comparability:** It follows the convention of prior work [1], allowing for a direct and fair comparison with existing methods.
> 2. **Motivation:** Our findings strongly support the reviewer's intuition. The key result that sparser verification (g\>1) is often optimal suggests that the conventional newline-delimited "thinking step" is indeed too fine-grained and not always meaningful.
>
> In essence, VG-Search with g\>1 acts as a simple but effective method to create larger, more substantial steps by grouping the atomic newline-delimited ones. We see our work as providing the motivation and the framework for future research into more sophisticated, semantic-aware verification steps.
>
> \[1\] Snell, Charlie Victor, et al. "Scaling LLM test-time compute optimally can be more effective than scaling parameters for reasoning." *The Thirteenth International Conference on Learning Representations*. 2025\.
>
> > Q2: Varying granularity dynamically
>
> We thank the reviewer for this thoughtful question. Dynamically adjusting the verification granularity $g$ during generation is a promising direction for future work, as mentioned in our Limitation section, and we are happy to share our thoughts and preliminary findings on how this could be implemented.
>
> Following your suggestion, we investigated the relationship between the verifier's confidence (in this case, its score on the current step) and the magnitude of the score change on the subsequent step.
>
> **Our results show a clear inverse correlation:** the higher the verifier's confidence in the current step, the smaller the expected score change for the next step.
>
> | Verifier Confidence | 0-0.2 | 0.2-0.4 | 0.4-0.6 | 0.6-0.8 | 0.8-1 |
> | :---- | ----: | ----: | ----: | ----: | ----: |
> | Avg. consecutive-step Δscore | 0.290 | 0.274 | 0.190 | 0.110 | 0.020 |
>
> Based on this finding, a dynamic strategy could be implemented as follows:
>
> * Use a **larger verification granularity ($g$\>1)** when verifier confidence is high, saving compute when the path is stable.
> * Use a **denser verification ($g$=1)** when confidence is low to catch potential errors immediately.
>
> While a full evaluation is an exciting direction for future work, we will include this preliminary analysis and discussion in the revised version. Thank you for helping us improve the paper.
>
> > Q3: Open-ended questions performance
>
> We focus on mathematical reasoning tasks in this paper to stay comparable with prior work \[1\] and because most open-source PRMs are trained primarily on MATH. To test the generalizability of our conclusions beyond this domain, we applied VG-Search to HumanEval using LLaMA-3.2-1B as the generator and Skywork-o1-1.5B-PRM as the verifier, using newline characters as step delimiters.
>
> |  | $g$=1 | $g$=2 | $g$=3 | $g$=4 |
> | :---- | :---- | :---- | :---- | :---- |
> | n=4 | 0.250 | **0.274** | 0.238 | 0.220 |
> | n=16 | **0.335** | 0.305 | 0.268 | 0.311 |
> | n=64 | 0.256 | **0.335** | 0.274 | 0.305 |
>
> These results reinforce our main conclusion: **$g$=1 is not always optimal**, and with increased compute budget, **sparser verification granularity tends to perform better**. This demonstrates the applicability of our findings to open-ended code generation tasks as well.
>
> \[1\] Snell, Charlie Victor, et al. "Scaling LLM test-time compute optimally can be more effective than scaling parameters for reasoning." *The Thirteenth International Conference on Learning Representations*. 2025\.
>
> > Q4: Change of the best granularity with a noisy PRM
>
> We thank the reviewer for this insightful question about how verifier quality affects the optimal granularity.
>
> To investigate this, we conducted a "worst-case" ablation study where the verifier was maximally "noisy and poorly calibrated"—it was a **purely random PRM** that assigned a random score to each step. We tested this with both our 1.5B and 7B generators.
>
> Results on MATH-500
>
> | n | Qwen-1.5B |  |  |  | Qwen-7B |  |  |  |
> | :---- | ----: | ----: | ----: | ----: | ----: | :---- | :---- | :---- |
> |  | **$g$=1** | **$g$=2** | **$g$=3** | **$g$=4** | **$g$=1** | **$g$=2** | **$g$=3** | **$g$=4** |
> | 4 | 71.4 | 70.8 | 68.4 | **71.8** | 74.8 | 76.2 | 78.2 | **78.6** |
> | 16 | 68.0 | 73.2 | **73.8** | 73.2 | 78.8 | 79.6 | 77.8 | **80.6** |
> | 64 | 76.0 | 78.6 | 78.2 | **79.4** | 80.6 | 82.0 | 84.0 | **85.0** |
> | 128 | 77.4 | 78.4 | 79.0 | **80.4** | 80.0 | 84.4 | **85.8** | 85.4 |
> | 256 | 78.8 | 79.8 | 79.2 | **81.0** | 80.6 | 85.0 | 86.4 | **86.4** |
>
> Results on AIME
>
> | n | Qwen-1.5B |  |  |  | Qwen-7B |  |  |  |
> | :---- | ----: | :---- | :---- | :---- | ----: | ----: | :---- | :---- |
> |  | **$g$=1** | **$g$=2** | **$g$=3** | **$g$=4** | **$g$=1** | **$g$=2** | **$g$=3** | **$g$=4** |
> | 4 | 8.9 | 5.6 | 8.9 | **11.1** | 10.0 | **10.0** | 7.8 | 8.9 |
> | 16 | 11.1 | **12.2** | 6.7 | 11.1 | 12.2 | **20.0** | 11.1 | 13.3 |
> | 64 | 6.7 | 8.9 | 11.1 | 11.1 | 12.2 | 15.6 | **17.8** | 16.7 |
> | 128 | 12.2 | **14.4** | 11.1 | 11.1 | 13.3 | 15.6 | 15.6 | **17.8** |
> | 256 | 12.2 | 10.0 | 13.3 | **13.3** | 16.7 | 16.7 | **17.8** | 16.7 |
>
> Our hypothesis was that a noisy verifier would favor sparser checks to avoid corrupting the search with bad guidance. The results strongly confirm this intuition. Across both models and datasets, when the verifier provides no useful signal, **the optimal strategy consistently shifts to sparser verification (g \> 1\)**.
>
> In essence, with an unreliable verifier, the best strategy is to trust the generator more and the verifier less. VG-Search with a larger g naturally adapts to this by shifting the search strategy away from fine-grained guidance and towards a more robust Best-of-N style exploration.

---

> ### Author Response · Authors · 2025-08-06
> **Thank you & Final Questions**
>
> Thank you for your time and for the thoughtful suggestions.
>
> As we approach the conclusion of the discussion period, could you please let us know if all your questions have been fully addressed?

---

### Official Review · Reviewer_SNq5 · 2025-07-03

**Clarity:** 3
**Significance:** 3
**Originality:** 3
**Rating:** 5
**Confidence:** 4

**Summary:**

This paper proposes Variable Granularity Search (VG-Search), a verifier-guided decoding algorithm that introduces a tunable granularity parameter g to control how often a reward model is queried.
It groups candidate sequences into segments, queries the verifier only at segment boundaries, and enables budget-aware tuning of verification frequency.
Experiments on MATH-500, AIME, and a validation subset shows that the optimal g depends on compute budget and generator-verifier strength; an adaptive strategy gains up to 3.1 pp accuracy over beam search and 3.6 pp over Best-of-N while cutting FLOPs by more than 52%.

**Questions:**

Q1: How would VG-Search handle tasks that supply only outcome-level natural-language feedback rather than deterministic unit tests?

**Ethical Concerns:**

["NO or VERY MINOR ethics concerns only"]

**Final Justification:**

Dear authors, Thanks for the response! After reading the rebuttal, I feel more confident in the strengths of the paper. I updated my score to 5.

**Limitations:**

yes

**Paper Formatting Concerns:**

The formatting is correct.

**Quality:**

3

**Strengths And Weaknesses:**

Strengths:
1. Proven scaling and effective extended framework. VG-Search generalizes two popular test time scaling methods with a single hyperparameter and supplies an explicit FLOPs model, enabling great trade-off analysis across budgets and model pairs.
2. Complete and comprehensive evaluation. The study sweeps g, branch factor, verifier size, task difficulty, and sample count, and validates both compute savings and accuracy gains with clear ablations and an adaptive policy that outperforms strong baselines on multiple datasets.
3. Detailed design analysis via experiments. The profiling further reveals that strong generators prefer sparse verification whereas weaker ones need dense checks, and that investing saved compute in a larger verifier is more beneficial than raising the branch factor.

Weaknesses:
1. This paper just names its checker “Skywork-o1 PRM” and states it keeps the last score, but it never tells its model size, training data, or prompt, making the speed and bias unclear.
2. No comparison with alternative verification paradigms. Although the related-work section recognises outcome-level verifiers, step-wise PRMs, generative PRMs and multi-verifier ensembles, the experiments test just a single discriminative PRM. Nevertheless, it's still positive to observe the test-time scaling on it.

---

> ### Author Rebuttal · Authors · 2025-07-29
>
> We thank the reviewer for the helpful feedback. We will clarify your doubts as follows.
>
> > Q1: More details on the PRM
>
> We thank the reviewer for the suggestion and will include the requested details in the revised version of the paper. As cited in lines 152–154, the Skywork-o1 PRM used in our experiments is taken from [1]. Specifically, we use the 1.5B and 7B variants: *Skywork-o1-Open-PRM-Qwen-2.5-1.5B* and *Skywork-o1-Open-PRM-Qwen-2.5-7B*. For detailed training information, please refer to [1]. Our prompt for Llama models follows [2], and our prompt for Qwen models is stated in line 494 of the appendix.
>
> We will ensure that the description of the PRM is made clearer and more prominent in the next version.
>
> [1] He, Jujie, et al. Skywork-O1 Open Series. 2024.
> [2] Edward Beeching, Lewis Tunstall, and Sasha Rush. Scaling test-time compute with open models. 2024.
>
> > Q2: Comparison with alternative verifiers
>
> To enrich our evaluation scope, we included additional experiments with multi-verifier ensemble [3], as mentioned by the reviewer. We used the Qwen-2.5-Math-1.5B as a generator and an ensemble of two different verifiers (Skywork-1.5B and RLHFLow’s LLAMA3.1-8B-PRM) by averaging their scores.
>
> | n | Optimal g (MATH-500) | Top Accuracy (MATH-500) | Optimal g (AIME) | Top Accuracy (AIME) |
> | :---- | :---- | :---- | :---- | :---- |
> | 4 | 1 | 76.0 | 4 | 13.33 |
> | 16 | 1 | 80.0 | 3 | 16.67 |
> | 64 | 4 | 81.0 | 4 | 18.89 |
> | 128 | 3 | 82.2 | 2 | 24.44 |
> | 256 | 4 | 81.2 | 4 | 23.33 |
>
> A complete table of results:
>
> | n | MATH-500 |  |  |  | AIME |  |  |  |
> | :---: | ----: | ----: | ----: | ----: | ----: | ----: | ----: | ----: |
> |  | **g \= 1** | **g \= 2** | **g \= 3** | **g \= 4** | **g \= 1** | **g \= 2** | **g \= 3** | **g \= 4** |
> | 4 | **76.0** | 73.2 | 71.6 | 69.6 | 12.22 | 8.89 | 11.11 | **13.33** |
> | 16 | **80.0** | 77.4 | 76.4 | 78.4 | 13.33 | 13.33 | **16.67** | 14.44 |
> | 64 | 79.4 | 80.4 | 80.8 | 81.0 | 16.67 | 16.67 | 16.67 | **18.89** |
> | 128 | 80.8 | **82.2** | **82.2** | 80.8 | 15.56 | **24.44** | 23.33 | 18.89 |
> | 256 | 80.2 | 80.0 | 80.2 | **81.2** | **23.33** | **23.33** | **23.33** | **23.33** |
>
> We observe that the results align with our conclusion: the optimal verification granularity $g$ is dynamic. While dense verification ($g$=1) is sometimes best at lower sample counts, sparser verification ($g$ > 1) consistently achieves top performance on both datasets as the compute budget (n) increases.
>
> Regarding other paradigms, outcome-level methods like ORM are not suitable for our experiments, which require step-level verification. As noted in the paper, we deferred comparison with generative PRMs due to their verification inefficiency under matched compute budgets and the limited availability of open-source models, though we agree this remains an important direction for future work.
>
> We will incorporate the new ensemble results and this discussion into the final manuscript to better address the scope of verification paradigms. Thank you for helping us strengthen the paper.
>
> [3] Lifshitz S, McIlraith SA, Du Y. Multi-agent verification: Scaling test-time compute with multiple verifiers. 2025.
>
> > Q3: How does VG-Search handle tasks that provide outcome-level feedback?
>
> For tasks with outcome-level natural language feedback, if a PRM can be trained on intermediate generation steps, VG-Search remains directly applicable. The main difference lies in selecting the final answer, as majority or weighted majority voting isn’t feasible due to the open-ended nature of outputs. A practical alternative is to select the candidate with the highest PRM score. To illustrate this, we evaluated VG-Search on HumanEval using LLaMA-3.2-1B as the generator and Skywork-o1-1.5B-PRM as the verifier, with newline characters serving as the step delimiter.
>
> |  | $g$=1 | $g$=2 | $g$=3 | $g$=4 |
> | :---- | :---- | :---- | :---- | :---- |
> | n=4 | 0.250 | **0.274** | 0.238 | 0.220 |
> | n=16 | **0.335** | 0.305 | 0.268 | 0.311 |
> | n=64 | 0.256 | **0.335** | 0.274 | 0.305 |
>
> These results support our main conclusion: **$g$=1 is not always optimal**, and with increased compute budget, **sparser verification granularity tends to perform better**. This proves the generalizability of our conclusion on optimal verification granularity to open-ended tasks.

---

> > ### Comment · Reviewer_SNq5 · 2025-08-07
> >
> > Dear authors, Thanks for the response! After reading the rebuttal, I feel more confident in the strengths of the paper. I updated my score to 5.

---

> ### Author Response · Authors · 2025-08-06
> **Thank You & Final Questions**
>
> Thank you once again for your time and the insightful feedback.
>
> As we near the end of the discussion period, may I kindly ask if our responses have fully addressed all of your questions?

---

### Author Response · Authors · 2025-08-04
**Thank you & Look forward to the discussion**

Dear Reviewers,

Thank you again for your helpful feedback and for the constructive suggestions. We have tried our best to address your comments with experiments and clarifications, and we hope our responses have resolved your main concerns.

As the discussion period comes to a close, we would be grateful for any acknowledgement or discussion from you. Please let us know if any questions remain, as we would be happy to provide further clarification.

Sincerely,

The Authors

---

### Note · Authors · 2025-08-12

Dear AC and reviewers,

We sincerely thank all reviewers for their time and insightful feedback, which helped us improve our paper.

We are encouraged that the reviewers found our work to address a **novel and significant** aspect of test-time scaling (**Reviewer vtyM, Reviewer jWiE**) , with a **valuable standalone contribution** (**Reviewer jWiE**) , a **sensible and effective method** (**Reviewer RT2j, Reviewer SNq5**) , and a **comprehensive evaluation** (**Reviewer SNq5**).

During the rebuttal, we were pleased to address the reviewers' main concerns with additional experiments and analysis:
* To address **Reviewer SNq5's** questions on experimental scope, we provided new results using a multi-verifier ensemble. We are grateful that the reviewer felt "more confident in the strengths of the paper" and subsequently raised the score.
* For **Reviewer RT2j**, we validated our claims at a larger scale with a **14B model** and added **multi-trial runs**. The reviewer confirmed these additions addressed their concerns.
* For **Reviewer jWiE**, we provided a detailed convergence analysis for our adaptive strategies and expanded our experiments to a full 2x2 grid of model pairings. We were delighted the reviewer acknowledged this discussion "will help strengthen the paper quite a bit".
* **Reviewer vtyM's** insightful questions prompted us to include a new ablation study on noisy verifiers, further strengthening the work.

We believe, as acknowledged by the reviewers, most questions and concerns have been resolved during rebuttal. Thank you for your consideration.

Best regards,

Authors

---

### Decision · Program_Chairs · 2025-09-17

**Decision:**

Accept (poster)

**Comment:**

This paper proposes Variable Granularity Search (VG-Search), a verifier-guided decoding algorithm that introduces a tunable granularity parameter g to control how often a reward model is queried. It groups candidate sequences into segments, queries the verifier only at segment boundaries, and enables budget-aware tuning of verification frequency.

All the reviewers were positive at the end about our paper and wanted to accept it. I concur with their decision.